# Paternal Perinatal Experiences during the COVID-19 Pandemic: A Framework Analysis of the Reddit Forum Predaddit

**DOI:** 10.3390/ijerph20054408

**Published:** 2023-03-01

**Authors:** Emily E. Cameron, Kaeley M. Simpson, Shayna K. Pierce, Kailey E. Penner, Alanna Beyak, Irlanda Gomez, John-Michael Bowes, Kristin A. Reynolds, Lianne M. Tomfohr-Madsen, Leslie E. Roos

**Affiliations:** 1Department of Psychology, University of Manitoba, Winnipeg, MB R3T 2N2, Canada; 2Department of Community Health Sciences, University of Manitoba, Winnipeg, MB R3E 0W2, Canada; 3Department of Psychiatry, University of Manitoba, Winnipeg, MB R3E 3N4, Canada; 4Children’s Hospital Research Institute of Manitoba, Winnipeg, MB R3E 0Z3, Canada; 5Faculty of Education, Educational and Counselling Psychology, and Special Education, The University of British Columbia, Vancouver, BC V6T 1Z4, Canada; 6Department of Pediatrics and Child Health, University of Manitoba, Winnipeg, MB R3A 1S1, Canada

**Keywords:** perinatal, paternal, mental health, online forum, COVID-19

## Abstract

During the COVID-19 pandemic, new parents were disproportionately affected by public health restrictions changing service accessibility and increasing stressors. However, minimal research has examined pandemic-related stressors and experiences of perinatal fathers in naturalistic anonymous settings. An important and novel way parents seek connection and information is through online forums, which increased during COVID-19. The current study qualitatively analyzed the experiences of perinatal fathers from September to December 2020 through the Framework Analytic Approach to identify unmet support needs during COVID-19 using the online forum predaddit on reddit. Five main themes in the thematic framework included forum use, COVID-19, psychosocial distress, family functioning, and child health and development, each with related subthemes. Findings highlight the utility of predaddit as a source of information for, and interactions of, fathers to inform mental health services. Overall, fathers used the forum to engage with other fathers during a time of social isolation and for support during the transition to parenthood. This manuscript highlights the unmet support needs of fathers during the perinatal period and the importance of including fathers in perinatal care, implementing routine perinatal mood screening for both parents, and developing programs to support fathers during this transition to promote family wellbeing.

## 1. Introduction

Becoming a parent is a significant, life-changing experience [1].

For new fathers, there is now a substantial body of literature that has underscored the increased risk for mental health concerns and parenting stress [2,3]. Yet, there is still a limited understanding of the topics of concern to fathers and the natural ways in which they seek information to support parenting and mental health. Notably, the stressors of becoming a father have been magnified by the COVID-19 pandemic due to public health restrictions that have affected daily family life and service access [4,5]. Emerging preliminary literature during the pandemic suggests a nearly 10-fold increase in paternal anxiety and depression [6] from pre-pandemic meta-analyses and systematic reviews (e.g., [2,3]). Parental depression during the pandemic has also been found to be the single most important predictor of lower-quality parenting [7], highlighting the significant need to address mental health concerns and related risk factors to prevent negative outcomes for children and families. However, few perinatal programs directly target fathers’ needs and no studies have identified the specific stressors of fathers during (and following) the pandemic, despite substantial changes to paternal involvement in the context of the COVID-19-related restrictions. Thus, a deeper understanding of the experiences of expectant fathers, especially within the context of COVID-19, is necessary to inform programs and interventions to ensure fathers and their families are supported during and after the pandemic.

Recent reviews on paternal mental health and stress during the transition to parenthood have underscored several risk and protective factors for perinatal adjustment. Risk factors include co-parenting conflict, parity, parenting stress, unplanned pregnancies, substance use, poor relationships with parents, and maternal depression [8,9]. Conversely, protective factors for positive adjustment include having strong social and spousal support, high relationship satisfaction, and positive relationships with their own parents [8,9]. Poor perinatal adjustment and mental health can negatively impact the quality of life for fathers, decrease marital satisfaction, alter the quality of father-child interactions, and have a detrimental impact on child development [10], resulting in negative and persistent outcomes for the whole family system without adequate intervention. The limited qualitative research on fathers’ psychological and support needs include the following: desires for father-specific support and information in prenatal care and courses; emotional support and assurance that they are not alone; to feel more included by healthcare professionals; and to be involved in the birth [1,11]. Yet, the landscape of family stressors has shifted in recent years.

During the COVID-19 pandemic, many of the stressors experienced by expectant and new fathers have been exacerbated due to global shutdowns and restrictions that limit in-person contacts. The COVID-19 pandemic has changed the standard medical practices of pregnancy, childbirth, maternal postpartum care, and infant pediatric care, with many countries (e.g., Europe, Australia, North America) restricting fathers’ access to perinatal medical appointments such as prenatal ultrasounds, routine appointments, the birth of their child, or even the NICU in an effort to reduce the spread of COVID-19 [5,12]. These changes are especially concerning for fathers, given that involvement in prenatal care and knowledge access are important factors that can contribute to perinatal mental wellbeing, social support, and adjustment [13]. Widespread physical distancing mandates and restrictions that prohibit seeing family and friends have also led to fathers reporting greater feelings of social isolation as a result of COVID-19 [5]. These factors combined create further risk for adverse perinatal psychological and adjustment outcomes. However, there is limited research on paternal perinatal experiences during the COVID-19 pandemic to support intervention efforts.

Online forums and social media are novel emerging areas of research to address these knowledge gaps through naturalistic observations of fathers’ interactions, which can be coded at a large scale using rigorous content-analytic approaches. This allows for the analysis of rich and anonymous information on the experiences of a large existing sample [14]. These forums offer candid information about perinatal experiences that are not as susceptible to response biases typical in laboratory research [14]. Father forums are also easily accessible and allow for people who share demographic characteristics, such as being an expectant father, to readily engage and converse with one another [15,16]. However, there is a dearth of research that has examined the content of forums of perinatal fathers and no research to date on how forums reflect fathers’ perinatal experiences during the COVID-19 pandemic.

Thus, the present research extends previous literature on the needs of fathers during the transition to parenthood through a qualitative analysis of an online forum for expectant fathers during the COVID-19 pandemic. The aims of the current research are twofold: (1) to investigate the perinatal experiences of men within the context of COVID-19, and (2) to identify the specific support needs and unmet needs of expectant fathers. We address these research questions through a qualitative analysis of an online forum to obtain naturalistic observations of social support during the transition to fatherhood. The ultimate goal of this work is to inform family-centered and father-tailored intervention and program development through evidence-based recommendations for therapeutic content and therapeutic components (e.g., peer support) to support fathers during the transition to parenthood.

## 2. Materials and Methods

This project adhered to the standards for reporting qualitative research (SRQR) checklist for qualitative studies [17]. The Framework Analytic Approach was employed for qualitative analyses of posts and comments from the selected forum. The Framework Analytic Approach allowed for establishing an a priori framework based on previous literature and our specific research questions, which were refined during the analysis phase to capture the full scope of the fathers’ perspectives. Such an approach involves reflexively and flexibly moving through five stages: familiarization with the data, identifying the thematic framework, indexing and sorting the data, reviewing extracts, mapping, and interpretation [18].

### 2.1. Forum Selection

Analyzing father forums is a valid method of collecting data, as demonstrated through previous forum analyses [19,20]. Reddit.com was chosen for the current forum analysis, given its popularity [21] and previously identified use for qualitative research [22]. This forum allowed for qualitative analysis of a specific subforum called predaddit “for men about to become fathers” [23], which was devoted to all topics for expectant fathers. Predaddit was created in 2011, contained over 42,600 members, and had 8.61 posts per day in November 2020, highlighting the popularity for and regular engagement of fathers on this forum during the period of data collection.

### 2.2. Data Collection

Two trained coders (KS, KP) extracted all posts and comments from predaddit between 1 September 2020 and 31 December 2020 into two Microsoft Excel documents to gain familiarity with the data. The timeframe was chosen to ensure that posts were within the context of the COVID-19 pandemic; specifically, this period was broadly representative of the second wave of the pandemic for many countries worldwide, in addition to a return to (or consideration of) in-person work and school for many parents [24]. Data collection included the date the post or comment was retrieved, the date the main post was created, the title of the main post, the username of the original poster, the content of the main post, the number of words in the main post, the number of upvotes (i.e., defined as a button for users to express whether a post contributes and is relevant or not to a subreddit, and/or whether they like the content or not), the number of comments, and all comment content. This process resulted in 792 posts with a combined 8011 comments. The significant number of posts and comments indicated that sampling saturation was likely in the a priori selected time period. Posts involving advertisements or spam and those not written in English were excluded. The Microsoft Excel sheets were then combined into one document representing all posts and comments and imported into NVivo 12 (QSR International) for analysis.

### 2.3. Initial Thematic Framework

Following familiarization with the data, the initial framework was created by all authors under the supervision of and final confirmation from senior authors (EC, KR, LR). The initial framework involved six main themes: psychosocial distress (subthemes: parental mental health, physical health, coping, role identity and masculinity, and adjustment and grief), family functioning (subthemes: resource insecurity, domestic conflict, co-parenting relationships, and maternal health and supporting the co-parent), child health and development (subthemes: child development, child physical health and medical concerns, family history concerns, and mention of miscarriage or stillbirth), parenting (subthemes: parenting self-efficacy and parental or pregnancy-related decision making), forum use/support interactions (subthemes: requesting advice or support, posts giving advice or support, posts announcing accomplishments, sharing information, general engagement, irrelevant, mother or other, and reaction to mother or other), and interpersonal functioning (social difficulty with family, friends, and social isolation or loneliness). Comments that mentioned COVID-19 were double-coded with the appropriate main and subthemes as well as a code for COVID-19.

Given the public nature of predaddit, a range of participant characteristics was not able to be determined for every poster. Within the initial framework, we added a main theme to capture the stage that fathers were within the perinatal period if mentioned in the post. Subthemes included pre-conception (e.g., wanting to get pregnant, wanting a child), pregnancy (defined as any trimester), and postpartum (defined as any time following childbirth). We also included a subtheme to capture whether this was the first pregnancy or a subsequent pregnancy (this code included families who were having another child, who experienced one or several miscarriages, stillbirth, etc.). Within our results, data coded under this timing theme was defined as participant characteristics, rather than as part of our final thematic framework. Posts were only coded to these subthemes if their post explicitly mentioned one of these characteristics. While only a small portion of the data gave a specific time period (e.g., 2-weeks postpartum), the majority of posts indicated generally that fathers were in the prenatal or immediate postpartum period (e.g., “But the pregnancy still doesn’t feel real to me”).

### 2.4. Coding and Rigor

The total 8803 extracted comments and posts were evenly divided amongst four trained coders. To assist with coding and coder training, the lead author and senior co-authors developed a list of themes and subthemes a priori, which was then modified over the course of data extraction to allow for additional relevant themes to be included. Summaries and examples of the themes and subthemes were provided to each coder. An initial training occurred where all four coders independently coded comments and posts to compare to the others’ coding. Discrepancies were discussed as a team under lead-author supervision until coders were in consistent agreement (i.e., >95%). Each coder then independently reviewed their designated comments and coded the posts and comments using the initial thematic framework. Each coder kept a coding journal to track coding questions, challenges, and resolutions. All challenges and questions were discussed during weekly team meetings under the supervision of the first author until a unanimous decision was reached. If needed, challenges and questions were extended to additional co-authors (KR, LR) for consensus. Additionally, as data were analyzed, all additions or adaptations to the thematic framework were discussed in such meetings. The mapping and interpretation stages were also addressed during the weekly team meetings by discussing emerging patterns and whether the full scope of the data mapped onto main and subthemes. An audit trail of all decisions made throughout the coding process was achieved through each coder keeping a detailed coding journal. Discussing challenges as a team ensured all coders had an equal understanding of the data and the thematic framework and facilitated reflexivity in the coding process. Such practices align with standards for ensuring rigorous results in qualitative studies [25] and are consistent with several high-quality qualitative studies [26,27].

## 3. Results

### 3.1. Participant Characteristics

Given the nature of the forum data, the characteristics of each participant could not be determined. Posts that explicitly included timing in the perinatal period were coded accordingly (Table 1). Of the 951 posts that included this information, posts were evenly distributed across trimesters, with relatively fewer posts in the postpartum or pre-conception periods; the majority of these posts were from first-time parents. The remaining posts in the analysis did not mention timing within the perinatal period or whether it was the user’s first or subsequent pregnancy.

### 3.2. Themes and Subthemes

Our analysis determined five main themes: forum use, COVID-19, psychosocial distress, family functioning, and child health and development. Each main theme was broken down to consist of subthemes. From the initial six-theme framework, the main theme of parenting and interpersonal functioning was merged to become a subtheme within the main theme of family functioning due to an overlap in content. Additionally, several subthemes were collapsed together due to overlapping content, including the following: child physical health concerns and family history concerns; COVID-19 adjustments and COVID-19-related emotions; co-parenting relationship and domestic conflict; accomplishment announcements and general engagement; coping and adjustment; and, irrelevant, mother or other, and reaction to mother or other.

The frequency of codes for each theme and subtheme in the analysis is displayed in Table 2. The final thematic framework is displayed in Figure 1. Quotes are presented in the text as numbers (e.g., #1) with corresponding quotes displayed in Table 3.

### 3.3. Main Theme of Forum Use (8304 Codes, 76.8% of All Coded Posts)

Fathers used the forum to connect and engage with other expectant fathers. This theme encompasses the ways in which fathers communicated, including the subthemes of requesting advice or support, giving advice or support, sharing information, and general engagement. Posts made by non-fathers or non-expectant fathers were coded under the subtheme of irrelevant.

#### 3.3.1. Advice or Support Request (352 Codes, 4.2%)

Fathers used the predaddit forum as a modality to receive support and advice from other expectant fathers. Some fathers’ posts included requests for general advice, with one father posting: “I’m excited as can be, but can’t help but feel incredibly nervous every time I look at my pregnant wife’s baby bump. Any advice for a pre-dad who is freaking out?”. Other users asked for more specific advice such as good books and podcasts for expectant fathers (#1), how to handle anxiety during pregnancy (#2), and must have items for once their baby is born (#3).

#### 3.3.2. Advice or Support Giving (2516 Codes, 30.3%)

Many fathers replied to posts by providing support and advice to other users. This advice often included sharing personal stories and tips based on what worked well for them during the challenging times of pregnancy. For example, one father said: “we swear by ginger ale for nausea,” and another shared advice about planning things in advance prior to the baby arriving (#4). In addition to advice, the forum community included users providing a vast amount of reassurance and support to expectant fathers on topics including parenting competency (#5), the neonatal intensive care unit (#6), and miscarriage (#7). These posts often included words of encouragement, sympathy, and reassurance to which original posters often thanked other users or expressed gratitude for the support.

#### 3.3.3. Sharing Information (318 Codes, 3.8%)

Within the forum, fathers shared websites, informational resources, and recommendations for supplies, books, and podcasts with other expectant fathers. Sometimes these were in response to users asking for these resources (#8); other instances included fathers sharing the information they had personally found helpful to the general forum community (#9). The majority of shared information was tools and resources pertaining to preparing to become a father.

#### 3.3.4. General Engagement (4550 Codes, 54.8%)

Posts coded to general engagement included engagement between users that was relevant to include in the analysis, but the content of these posts did not fit any of the themes within the present framework. These posts included fathers generally engaging and communicating with one another about topics such as movies, sports, hobbies, and interests. In addition, many predaddit users posted in the forum when their child was been born stating that they had “graduated” from being a “pre-dad,” to which other users replied with congratulatory messages; all of which were coded as general engagement. User comments that were neutral in nature (e.g., “I understand”) as opposed to direct advice or support giving were also coded as general engagement.

#### 3.3.5. Irrelevant (568 Codes, 6.8%)

Posts and comments were coded as irrelevant if they were posted by an individual who identified as anything other than an expectant father or father (e.g., mothers, moderators of the forum). Deleted or unfinished posts and comments were also coded as irrelevant.

### 3.4. Main Theme of COVID-19 (543 Codes, 5.0% of All Coded Posts)

Fathers discussed COVID-19 and how they have been impacted by the pandemic. Conversations about COVID-19 included discussions about the subthemes of adjustments and emotions, as well as service access disruptions.

#### 3.4.1. COVID-19 Adjustments and Emotions (287 Codes, 52.9%)

Users shared experiences of how COVID-19 led to adjustments to their current lifestyles. These adjustments included self-isolation or being away from their partner if they had or were in close contact with someone with COVID-19, not seeing family and friends, and participating in fewer social activities (e.g., participating in sports, going out to restaurants). Fathers shared that they made these decisions to keep their child safe (#10). Making these decisions often caused stress for families in addition to the stress and worry they experienced about their child or partner contracting COVID-19. One user shared: “We are terrified of the birthing process. With COVID in the world, our fears have skyrocketed.” Users also shared the emotions they were experiencing due to COVID-19. Fathers expressed how they felt isolated due to COVID-19 restrictions and not being able to see friends and family, as well as struggling to feel positive emotions in the context of the pandemic (#11).

#### 3.4.2. Service Access Use or Disruption (256 Codes, 47.1%)

Many posts about COVID-19 included fathers talking about access to perinatal services, including the inclusion of fathers at prenatal appointments and attendance during their partner’s delivery. Fathers posted about the various restrictions due to the pandemic that limited their ability to be fully involved throughout their partner’s pregnancy. Users shared not being allowed to attend doctor’s appointments (#12) or ultrasounds (#13) and being prohibited from the hospital delivery room (#14). These restrictions were extremely disappointing to and difficult for fathers, especially first-time fathers. One father shared: “I haven’t been allowed to attend any scan yet and it’s killing me.” Many fathers explained that these restrictions contributed to the pregnancy not feeling real for them (#15), and they felt that they were unable to truly support their partners as they could not attend important appointments (#16). Fathers provided advice to each other on how to navigate these restrictions and challenges, with some suggesting having their partner connect with them virtually (e.g., Facetime) while at the appointment and others suggesting attending privately owned companies for ultrasounds, noting they generally have less-strict restrictions than public hospitals.

### 3.5. Main Theme of Psychosocial Distress (745 Codes, 6.9% of All Coded Posts)

Fathers used the forum to discuss general challenges and experiences of psychosocial distress related to the transition to parenthood. These posts included discussions about the subthemes of coping and adjustment, paternal mental health, physical wellbeing, and role identity and masculinity.

#### 3.5.1. Coping and Adjustment (249 Codes, 33.4%)

Users discussed how they were adjusting to and coping with the new experience and emotions of becoming a father. Many shared sentiments of happiness and excitement (#17), while others shared feelings of nervousness and stressors they were experiencing throughout the perinatal period (#18). These posts often included fathers seeking reassurance that their feelings were normal and asking if others had similar experiences. In response to others, forum users replied that experiencing feelings of worry, stress, and anxiety during pregnancy was “totally normal.” For example, one user said: “It took me about 7 months to really get excited. I’m 4.5 months in now and couldn’t be happier. It’s a big change and takes a lot to wrap your head around but it’ll come.” Posts coded to this subtheme also included discussions surrounding grief, mainly due to experiences of miscarriage and how fathers were adjusting to and coping with this loss (#19).

#### 3.5.2. Paternal Mental Health (251 Codes, 33.7%)

Within the forum, fathers posted about their personal mental health, often sharing stories about their mental wellbeing throughout their partner’s pregnancy. Many expectant fathers reported experiencing anxiety about their partner’s labor (#20), about being a good father (#21), and pregnancy-related anxiety about their baby’s or partner’s health (#22). Other discussions about mental health included posts about expectant fathers experiencing feelings of depression and stress (#23). In response, other fathers shared information reaffirming the importance and validity of mental health challenges of fathers, with one user posting: “Prenatal depression and anxiety is a thing and can affect both mom AND dad. Same thing goes for postpartum.” Discussions of mental health also included posts from fathers about positive wellbeing, with some fathers sharing feelings of happiness, optimism, gratitude, and excitement. In one post, a user stated: “I’m filled with awe and happiness and gratitude”.

#### 3.5.3. Physical Wellbeing (106 Codes, 14.2%)

In addition to mental health, fathers posted asking for advice on how to maintain good physical wellbeing and sharing how their physical health had been impacted since becoming a father or expectant father. Users discussed the importance of exercising, eating healthy, and prioritizing their physical wellbeing (#24). Other posts included conversations around lack of sleep and fatigue, such as “I’m probably only getting three good night’s sleep per week” and “god do I miss sleeping longer than 2 hours.” Fathers experiencing a subsequent pregnancy often provided advice to first-time fathers emphasizing a need for fathers to take care of themselves (#25). Additionally, users posted about strategies to maintain good physical wellbeing during this time (#26).

#### 3.5.4. Role Identity and Masculinity (139 Codes, 18.7%)

Identity was discussed both within the context of becoming a father and masculinity. Some fathers expressed concerns about not “feeling” like they were going to be a dad and the pregnancy not feeling “real” to them. Many fathers explained that it took a while for the idea that they were going to be a father to “truly sink in.” Some fathers explained that these feelings may stem from the lack of inclusion of fathers during the perinatal period. Fathers reported feeling left out during prenatal courses (#27); they also noted the limited access to and/or amount of resources designed specifically with fathers in mind, including that “all the books and websites just talk about ‘mom and baby.’” Some fathers shared that the resources and information available to dads often appeared “patronizing” and presumed that fathers would not be invested or interested in fatherhood (#28). Fathers were disapproving of these types of messages with many users sharing their commitment to being fully and actively involved through all components of fatherhood.

### 3.6. Main Theme of Family Functioning (850 Codes, 7.9% of All Coded Posts)

Expectant fathers used the forum to discuss items related to various components of family functioning such as the subthemes of co-parenting relationship, parenting, maternal health and supporting their partner, social difficulties, and resource insecurities. 

#### 3.6.1. Co-Parenting Relationship (210 Codes, 24.7%)

Fathers used the forum to discuss their relationship with their partner, and changes to the relationship as a result of becoming expectant parents. Some users discussed less intimacy with their partner (#29), changes to household division of labor (#30), and shared discussions they were having with their partner about parenting styles and strategies. Many users shared how their emotional connection with their partner had changed since becoming pregnant; some had become closer and stronger (#31), while others noted their relationship had become colder and more distant (#32). Within these posts, users expressed their partners were more irritable and had frequent “mood swings” during pregnancy (#33) to which other fathers reassured them that these emotions were normal and likely a result of pregnancy hormones (#34).

#### 3.6.2. Parenting (243 Codes, 28.6%)

Posts coded to the subtheme of parenting included seeking advice on pregnancy and/or parenting related decision-making and expressing thoughts regarding parenting efficacy. Users sought advice and posted about making decisions such as giving birth in a hospital or at home (#35), infertility treatments (#36), and deciding on COVID-19 precautions to protect their newborn from infection (#37). In response to these posts other users shared their own stories, decisions they made during the perinatal period, and advice to other expectant fathers. Posters also discussed their self-efficacy in relation to parenting with many stating that they hoped they would be a good father, and others asking for advice on how to be a good parent and provide for their children (#38).

#### 3.6.3. Maternal Health and Supporting the Other Parent (526 Codes, 61.9%)

Many users shared stories about the health of their partner during the perinatal period. Maternal health posts included discussions about preeclampsia, maternal fatigue, and morning sickness (#39). Users also posted about the health and wellbeing of mothers during labor, with some noting quick and smooth labor experiences (#40) and others sharing their difficult labor experiences (#41). Within the posts where fathers shared the health challenges their partner was experiencing, they often discussed ways in which they were supporting their partner through these times (#42). These posts also frequently included users asking for advice on how they could be a better source of support to their partners throughout pregnancy and labor (#43). One father said: “I want to do something special for my wife after she goes through the exhausting and painful process of giving birth to our daughter. Got any ideas?”.

#### 3.6.4. Social Difficulties (91 Codes, 10.7%)

Fathers expressed social difficulties with family mainly surrounding managing visits between the parents’ extended family and their newborn within the context of COVID-19. Users shared that they have decided to either allow no visitors to see their child to protect their child from the risk of infection, while others shared the precautions they had set in place for their family members visiting such as “masks on” and “hands washed” (#44). One father said: “If they don’t quarantine for two weeks—no one sees my 3 month old till a vaccine is out.” Other fathers shared that they would not be attending family functions like Christmas gatherings and family dinners because of COVID-19 (#45). Fathers expressed that these decisions caused some disappointment to their extended family but emphasized that protecting their child’s health and wellbeing was a priority.

#### 3.6.5. Resource Insecurity (113 Codes, 13.3%)

Fathers used the forum to discuss expenses incurred and the lack of resources during the perinatal period. Users shared the costs of prenatal appointments and childbirth with one father noting: “We spent about $8500 on the pregnancy and birth of our daughter above what our health insurance paid for. I’m a small business owner and our healthcare options are limited and expensive.” In response, fathers shared similar stories about how much pregnancy cost them, how it was challenging to “make ends meet” without insurance coverage, costs of infertility treatments, the high prices of ultrasounds and blood tests, and limited income during parental leaves from work. Fathers also noted the costs that arose in preparing for a child in pregnancy and post-childbirth, including purchasing car seats and paying for childcare. Notably, many fathers expressed that their financial stress was magnified due to the COVID-19 pandemic with fathers posting concerns about the “COVID job market” (#46). When fathers expressed feelings of financial stress, others replied with support and reassurance; one father stated: “If you have a partner who loves you and a healthy child on the way you sound like a wealthy man to me”.

### 3.7. Main Theme of Child Health and Development (369 Codes, 3.4% of All Coded Posts)

Fathers expressed concerns and sought advice on the physical health of their child, and discussed various concerns they had about the development of their baby. Fathers also discussed experiences of miscarriage and looked for advice and support from other fathers.

#### 3.7.1. Child Development and Health Concerns (226 Codes, 61.0%)

Fathers shared concerns about the physical health and development of their child in utero. Some prenatal developmental concerns included finding out the fetus was measuring small for the gestational age (#47) or that their child may have a disability or genetic disorder (#48). Within these posts fathers often asked if others had experienced something similar and what the outcome had been. One father asked, “They saw some concerning bright spots in his bowel that may indicate a chromosomal disorder. I know I will love my child no matter what issues he may have, but like any dad, I want them to be as few as possible. Have any of you had similar experiences?”. Fathers also discussed family medical or genetic history concerns that posed greater risk for their child being diagnosed with a disability or medical condition (#48).

#### 3.7.2. Miscarriage (145 Codes, 39.3%)

Many fathers shared that they and their partner experienced a miscarriage. Within these posts fathers explained their emotional experience, including depression, sadness, and disappointment. Fathers sought support from others during this challenging time with one father posting: “Hoping you can provide me with some advice/words of encouragement because I’m struggling right now!”. Others responded with their own experiences of miscarriage and provided support and encouragement to the couple to try again once they were ready (#49). Discussions normalizing the frequency of miscarriages were also prevalent within the forum (#50).

## 4. Discussion

This study is the first to use a qualitative approach to analyze the experiences and stressors of perinatal fathers during the COVID-19 pandemic. Findings from the current study highlight the popularity and utility of the predaddit forum as a source of information for and interactions of fathers, given that there were a total of 8803 posts and comments across themes. The main themes that arose in the qualitative analysis included forum use, COVID-19, psychosocial distress, family functioning, and child health and development. Fathers used the forum most frequently for general engagement and to provide advice and support to each other. Additional subthemes that were most frequently coded included discussions about maternal health and supporting the other parent, requesting advice or support, and sharing information.

Little information is available in the extant literature on fathers’ functioning or needs during and post-pandemic. However, the findings of the current study are consistent with emerging quantitative literature by our group and others that suggest the pandemic is having an adverse effect on some fathers due to COVID-19-related factors, including increased financial stressors [6], service disruption or exclusion, and social isolation [5]. Within the current study, uncertainty about COVID-19 and its impact on the perinatal experience led to increased self-reported fears and anxieties. Fathers also highlighted the effect of social isolation during a transition that benefits greatly from the social support of family and friends to adjust to parenthood. A notable consideration was the exclusion many fathers felt from the perinatal process, given restrictions on attendance at routine medical appointments, ultrasounds, and the birthing experience. These experiences were reported to lead to significant emotional responses and feelings of unpreparedness and disconnection from the growing child, which are associated with increased risk for fathers to experience mental health concerns during the transition to parenthood [28]. These findings are also consistent with the previous pre-pandemic qualitative literature that has reported fathers’ desire to be involved in and knowledgeable about pregnancy and childbirth [29,30]. Research suggests that fathers who are included in antenatal care and provided with perinatal resources and education experience better health and wellbeing; the inclusion of fathers similarly improves maternal and infant health [31]. These findings highlight potential mechanisms that may explain in part the increase in mental health concerns for fathers during the pandemic. Furthermore, they may foreshadow a persistent increase in fathers’ mental health concerns without adequate intervention and inclusion during and following the pandemic.

Consistent with previous quantitative [2] and qualitative literature [32,33,34], fathers acknowledged many difficulties faced during the perinatal period including mental health challenges, difficulties coping with stressors, and changes to the marital relationship. Fathers also used the forum to discuss fears and concerns around their child’s physical health and development in the womb, as well as seek support and comfort following difficult news (e.g., miscarriage). Importantly, fathers often responded to these posts with support and understanding. While paternal perinatal depression is an emerging area of research and is currently not well recognized in routine clinical practice [35], fathers on the forum commonly validated other fathers’ experiences, noting that mental health and adjustment during the transition are very important, and postpartum depression is a “thing” for fathers [32,33]. These findings are consistent with previously published themes for expectant fathers of a need for access to father-inclusive professional support and parental education [32,33]. However, the needs of mothers are often prioritized by professional staff, family members, and friends, leaving fathers to feel as though their own concerns and needs are neglected or unimportant [34]. Many prenatal courses indirectly exclude fathers’ specific needs through the content being tailored specifically to mothers’ needs, resulting in the normalization of lack of support and exclusion of fathers during the prenatal period [13]. Clearly, fathers are in need of, and actively seeking, connection and support regarding perinatal mental health challenges.

Fathers also often identified two important dyadic considerations. First, fathers discussed their role as a support for mothers during the transition to parenthood, which has been extensively considered in the research that has incorporated fathers into interventions to support maternal wellbeing [36]. However, these concerns primarily revolved around maternal health, ranging from standard pregnancy symptoms (e.g., nausea, fatigue) to more serious health considerations (e.g., preeclampsia), highlighting the impact that even typical pregnancy experiences have on both partners. Second, fathers used the forum to discuss parenting and the co-parent relationship, including the changes to the relationship brought on as a result of becoming a parent. Fathers discussed the changes to the intimate relationship and connection to their partner and the related stress, which is consistent with research that shows actor-partner effects of the marital relationship and spouse-related stress on prenatal depression [37]. Fathers in the forum often normalized typical experiences (e.g., “mood swings” related to pregnancy hormones), which may help to reduce stress in fathers; however, future research should consider the utility of normalizing typical dyadic experiences for fathers to reduce the risk for prenatal and postpartum mental health concerns.

The main theme of forum use underscored the expressed need of fathers to engage with other fathers, ask and offer support and advice to their peers, and share in the perinatal experience in a safe, anonymous space. Fathers consistently reported feeling supported by other fathers and stated the value of the forum. This finding extends previous qualitative literature that has reported fathers’ desire for services and programs tailored towards fathers specifically [32,33].

### 4.1. Practical and Clinical Implications

These findings can inform the development of evidence-based services that are geared specifically towards fathers to address unmet support needs. Recent research has demonstrated that a substantial number of individuals prefer online supports, including forums, for disclosing and discussing personal problems [38]. For fathers specifically, access to internet or app-based information has been evaluated in recent years (e.g., [39,40]). These programs have proven to be especially important during the pandemic with the reduction of in-person service access [41]. Our results suggest that fathers would also benefit from services that address increased stress, anxiety, and depression during the pandemic [6]; however, given the substantial use and utility of the predaddit forum, fathers may additionally benefit from programs that encourage anonymous, clinician-mediated interactions (e.g., forums) to promote open discussion and disclosure of personal concerns. The current study has additionally demonstrated the positivity and fulfillment expressed by many fathers through hearing perspectives and advice directly from fathers with lived experience.

### 4.2. Strengths and Limitations

Given the anonymous nature of the forum and posts, the specific demographics of the users could not be determined. When possible, sociodemographic information was extracted (e.g., gestational age, parity); however, the generalizability of the findings based on important sociodemographic indicators (e.g., income, education, parental age, country of residence, ethnicity) may be limited. These factors likely impact the main themes and subthemes discussed in the forum. Further, the length of posts and comments within a thread varied greatly, and the coding scheme was based on and limited to the information provided in these comments during the specified time period. Given that the incidence rates of COVID-19 infection in the community and related public health guidelines varied significantly across locations and time periods during the pandemic, the generalizability of these concerns may reflect a snapshot of information during the period of data collection. Nonetheless, the findings appear to be consistent with pre-pandemic themes identified by researchers in quantitative studies and by fathers directly in qualitative studies, suggesting that these themes are robust. Furthermore, this study adds to the emerging paternal literature to include the COVID-19-related needs and experiences of fathers during the transition to parenthood, which is relevant during and post-pandemic as services continue to be impacted and family health remains a concern for many. Lastly, this study demonstrated a strength in advancing qualitative published literature that has previously focused on qualitative interviews, which are limited in a number of ways. Such methods do not allow for anonymity and thus may be vulnerable to social desirability biases [42,43]. Individual data collection does not allow for the examination of how fathers interact with one another. These methods are also limited in their sample size due to methodological restraints on collecting qualitative data directly from large samples. As such, the current study provides an important naturalistic account of the fatherhood experience in a large sample that is not susceptible to social desirability biases.

## 5. Conclusions

The wellbeing of fathers is integral to the successful transition to parenthood, providing support for new mothers, and in promoting healthy child development. In addition to the expected challenges fathers face during the perinatal period, they have also been adversely affected by the COVID-19 pandemic. Now more than ever there is a significant need to build on research that understands fathers’ unique needs. This study is the first to describe those needs in a perinatal paternal population and highlight the immense utility of including peer support in interventions for fathers. Additional key recommendations across study findings include incorporating fathers in perinatal care, implementing routine perinatal mood screening for both parents, and developing programs to support fathers during this transition. A family-centered and father-inclusive approach to the perinatal period is key to promoting child development and long-term family wellbeing. 

## Figures and Tables

**Figure 1 ijerph-20-04408-f001:**
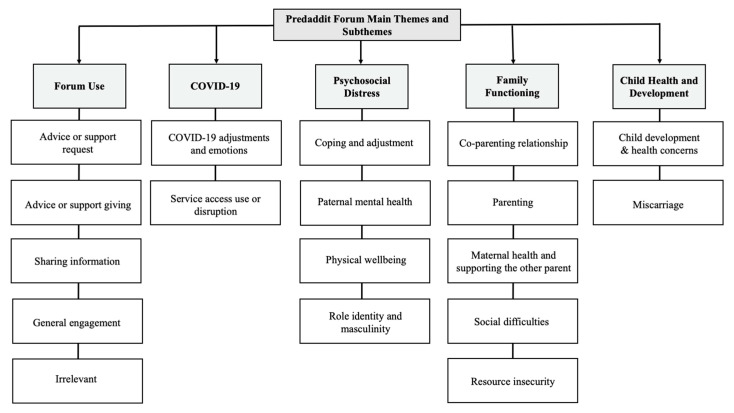
Final framework of main themes and subthemes.

**Table 1 ijerph-20-04408-t001:** Prenatal characteristics of the sample.

ALL CODES	TOTAL
**Timing in Perinatal Period**	**1128**
First Trimester (conception to 12 weeks)	276
Second Trimester (week 13 through 28)	235
Third Trimester (week 29 to end of pregnancy)	215
Pre-conception (wanting to have a child)	7
Postpartum	218
**Father Demographics**	
First-time father	134
Subsequent pregnancy	38

**Table 2 ijerph-20-04408-t002:** Frequencies of codes.

ALL CODES	TOTAL
**Forum Use**	**8304**
Advice or Support Request	352
Advice or Support Giving	2516
Sharing Information	318
General Engagement	4550
Irrelevant	568
**COVID-19**	**543**
COVID-19 Adjustments and Emotions	287
Service Access Use or Disruption	256
**Psychosocial Distress**	**745**
Coping and Adjustment	249
Paternal Mental Health	251
Physical Wellbeing	106
Role Identity and Masculinity	139
**Family Functioning**	**850**
Co-Parenting Relationships	210
Parenting	243
Maternal Health and Supporting the Other Parent	526
Social Difficulties	91
Resource Insecurity	113
**Child Health and Development**	**369**
Child Health Concerns and Development	226
Miscarriage	145

**Table 3 ijerph-20-04408-t003:** Supporting quotes for each main theme and subtheme.

Main Theme	Subtheme	Example	In-Text Identifier
**Main Theme 1: Forum Use**	**Advice or Support Request**	Hey all! Been lurking for a couple weeks and was wondering if there were any book recommendations for dads to be. Or books that you have read that you highly recommend. Thanks in advance!	1
		We have our anatomy scan next week, and while we already paid to find out the gender separately from the hospital I am filled with anxiety and fear that when we get the scan there will be no heartbeat. I am normally an anxious person so this is killing me. Does anyone else have these fears and how do you cope with them? If you had these fears did everything turn out okay?	2
		Anyone have a good link to anything that has must have lists for new parents?	3
	**Advice or Support Giving**	To make things easier on yourself, make a plan; even if you’re still at 9 weeks, check what a baby needs, check on furniture, decide when to start buying things.	4
		Just remember to breathe! You got this. The fact that you’re stressed shows how loving and caring you are. You are already an amazing dad.	5
		You can do this daddy. All good vibes and prayers for your family here. This little guy is a fighter and he will pull through!	6
		I am deeply sorry you and your partner are experiencing such a tragic loss. I hope your pain lessens each day and you both find joy in the near future.	7
	**Sharing Information**	Check out Mommy Labor Nurse (she has a website, Instagram account, and podcast). She’s a labor and delivery nurse and has online classes for childbirth prep as well as newborn care.	8
		I’m reading The Mayo Clinic Guide to Your Baby’s First Years and finding it does a great overview of a lot of different areas, including sleep.	9
**Main Theme 2: COVID-19**	**COVID-19 Adjustments and Emotions**	Facetime only. No visits. Ours was born on 20th October. Family understands and respects or they don’t but I personally would have zero issues hurting someone’s feeling if it meant keeping my child safe.	10
		Haven’t seen family in months. Barely saw friends this year. It’s hard to get excited by anything. Even the virtual baby announcements and now Xmas just doesn’t feel right.	11
	**Service Access Use or Disruption**	First doctor’s appointment is next Tuesday, and I’m not invited due to COVID. It sucks.	12
		Because of COVID, I’ve waited in the parking lot/garage of the hospital during every appointment and ultrasound. I haven’t met the OB doc in person. I end up seeing ultrasound pictures after the fact.	13
		In the early lockdown fathers were barred from the delivery room as well.	14
		But the pregnancy still doesn’t feel real to me. Because of COVID	15
		It felt like these people were robbing us of a special moment so we got really pushy until they let us facetime. Really annoying considering they let bars and all these other places open up but fathers can’t support their spouses. What sort of dumb nonsense is that?	16
**Main Theme 3: Psychosocial Distress**	**Coping and Adjustment**	It took me about 7 mo to really get excited. I’m 4.5 mo in now and couldn’t be happier. It’s a big change and takes a lot to wrap your head around but it’ll come.	17
		We are are both immigrants in New Zealand with no family and very little close friends. We found out that we are having twins. We were really happy one second and really scared the other. The plan was that my wife’s parents would come during delivery and help us with the babies. But due to border closures they can’t now. Now in a foreign country we have no support with two little babies on the way. I am a grad student doing my PhD. So I can only take a month off to help out. This has led me to stress more about the birth of my first and probably only children. More stress than enjoyment.	18
		We unexpectedly lost our little girl around 22 weeks a couple of years ago. We got some answers, the umbilical cord had gotten a pinch in it, but we didn’t feel satisfied with that at the time. With a lot of time spent at fertility specialist and just worrying every week, we finally were able to welcome our little boy just last week. Don’t lose hope, be there for each[other], but most importantly don’t let any anger or grief dwell within.	19
	**Parental Mental Health**	I’m having some anxiety about being in the delivery room. I want to be supportive of my wife and don’t want her to sense my anxiety. All I can think about is having a panic attack while she’s screaming in the delivery room.	20
		I’m embarking on a journey deep within myself to face my fears about becoming a father and building a family with the woman of my dreams. I did not have a positive experience with my own father, and didn’t have a super secure family life growing up, either. I’m wondering if anyone here (with similar experiences/anxieties) can recommend any resources, exercises or other help as I explore this path for myself and prepare to be the man I know I can be for my partner and our future horde of child geniuses. Thanks all!	21
		I’m worrying about a lot of things. What if my wife’s receding first trimester symptoms mean somethings wrong? What if we lose this baby?	22
		We are at 20 week mark next week too but I am highly stressed and maybe heading towards depression.	23
	**Physical Wellbeing**	At a minimum: exercise, improve your eating habits over the next few months. Make time for yourself to take care of yourself to be there, mentally and physically, for the long haul!	24
		Try to sleep as best as possible, eat as clean as possible, move as much as possible. It’s definitely a tough time in the weight gain department.	25
		I’ve been really focused on better self care though, trying to prioritize my hydration, a physical activity each day, and started a bedtime routine and set a bed time. I’ve never been one to prioritize myself before but I finally figured if I don’t I’ll run myself in to a hole and won’t be any good to anyone like that.	26
	**Role Identity and Masculinity**	I’m kind of coming around to the conclusion that ‘left out’ is pretty much par for the course for the non-birth partner.	27
		The book [is] a little patronizing towards dads as well, and written like we’re all beer drinking, football-watching couch potatoes that wouldn’t otherwise be interested in fatherhood.	28
**Main Theme 4: Family Functioning**	**Co-Parenting Relationship**	My wife is almost 20 weeks and we haven’t been intimate since we found out she was pregnant. Completely understandable reasons. She has not had an easy time and has not at all been in the mood. I’m doing a lot around the house, cooking, cleaning, bigger projects, etc.	29
		Things might not be the same for you but my wife has been extra tired since it all started. Being able to take a nap in the afternoon has helped her a lot. I was pretty amazed at how naturally I took over almost all chores and housework	30
		I have been much more emotional than usual. I find myself wanting to spend every spare minute that I have with my wife, more than I used to before.	31
		She used to be loving and affectionate but now she’s very cold and distant. Almost like I’m a burden on her. This all started about 4 weeks ago and there haven’t been many good moments between us since then. I understand her body is going through a lot right now and she is expectedly exhausted but it seems like she’s lost all feelings for me. I’ve tried talking to her about it but she just blows it off and never wants to talk.	32
		I know mood swings and irritability are certainly big parts of the pregnancy, but I am really struggling with it. She is becoming very mean and I’m starting to take it personally.	33
		Take some (or most) of the things my wife says with a grain (or pound) of salt, hormones can be a powerful thing.	34
	**Parenting**	Soooo my only options are probably a water birth in a birthing centre or even a home birth. Tbh I know very little about either so can anyone who has had experience of their partner delivering a baby in a water or home setting please share their experience of it so I can have some idea of what to expect?	35
		Just out of curiosity, why didn’t you move onto IVF? Our fertility doc told us that after ~3 IUI’s the chances of it working go way down so you might as well shift focus to IVF.	36
		Facetime only. No visits. Ours was born on 20th October. Family understands and respects or they don’t but I personally would have zero issues hurting someone’s feeling if it meant keeping my child safe.	37
		First time dad here. Just a year ago I was unmarried and living a carefree life. It’s been quite a ride since getting married. Added responsibilities and getting the news of becoming a father in May. Last year has transformed me into a responsible man. Wife’s EDD is 11 February. Just wanted to share this with you all. I hope I will be a good dad. Relying on this awesome sub reddit for useful advice from you all. Stay safe and healthy everyone.	38
	**Maternal Health and Supporting the Other Parent**	Morning sickness. Yeah seriously. My wife and I wondered why is it called ‘morning sickness’ when it seemed more like ‘all day sickness.’	39
		After 5 min of pushing, my daughter was born at 4:14 am. Now my wife can spend the next 18 years telling her how she gave birth with no epidural like a champ.	40
		My wife’s labor ended up being brutal. 46 h, followed by an emergency C-Section. Not a lot I could have done differently—except there was a nurse who was a complete BITCH to us. She was mean, and treated my wife rougher than she needed. I should have stepped in earlier—instead I waited for the end of her shift, and asked that she not be reassigned to us. At the first sign of trouble, I should have just gone to the doctor. It’s your job to make sure your partner’s wishes are respected. Y’all probably have some idea of what you want, but the important thing is that 2 healthy people go in, and three healthy people go out.	41
		My wife is 37 weeks. She lost her mucus plug a few days ago and her midwives think it could be any day now. Up until now I’ve been the nervous one, but it seems like we’ve swapped and she is having to work hard to feel ready. Any favorite encouraging phrases you all like to use to empower your S.O.?	42
	**Social Difficulties**	We were pretty transparent with everyone when it came to visiting. Masks on. Hands washed etc. Other than that, we don’t restrict them too much. If they refuse to follow those simple instructions, then they don’t get to meet the baby. Being a parent means being stern about things like this and you’re not going to be wrong for looking out for your family.	43
		We will not be attending any of the larger family Christmas gatherings. Thanksgiving dinner and Christmas will just be the same 7 other people we are closest to. I know extended family will be disappointed that they won’t get to meet my daughter but they’re going to have to wait until things calm down. We’re basically only taking her out of the house for doctors’ appointments. My cousins baby shower is scheduled for 4th December and the county will more than likely be in full lockdown by then. So that won’t be happening. It’s getting pretty bad.	44
	**Resource Insecurity**	Hopefully I’m doing enough for mom and baby, but I know I’ve been awake late because of worry about the COVID job market.	45
**Main Theme 5: Child Health and Development**	**Child Development and Health Concerns**	The baby would be around 7.5/8 weeks along but the ultrasound didn’t show that. There was no heartbeat. The doctor said the baby only measured at about 6 weeks 1 day, give or take. She said there are two reasons here. 1 is what is called a late ovulation (?) where the fertilization takes longer than it should which puts the growth behind. If that’s the case then the baby should be just fine, just a week or two behind. The reason, and the one we are obviously so worried about, is that the fetus was miscarried but just hasn’t shown any symptoms of that yet.	46
		We had the one test (I’m completely blanking on the name) where the baby’s head/spine was measured for the space between it and the cervix. We were told she was 30% likely to have a genetic disorder.	47
		My son had a 50% chance of basically being blind due to a X chromosome disorder we weren’t aware of. Luckily we dodged that and he’s fine.	48
	**Miscarriage**	Miscarriages are a lot more common than you think. I know it hurts right now, just don’t lose hope, heal with your spouse and try again when you are both comfortable!	49
		I understand that a miscarriage is a very sizeable possibility. She has the same outlook as me, and part of why we’re telling friends immediately is that we both agree that people should normalize talk of miscarriage.	50

## Data Availability

Available upon request.

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
