# Peer review of "Paternal Perinatal Experiences during the COVID-19 Pandemic: A Framework Analysis of the Reddit Forum Predaddit"

_ijerph, 2023, doi:10.3390/ijerph20054408_

Round 1

Reviewer 1 Report

This is an interesting article that discusses a topic that is widely ignored, i.e., paternal mental health. The authors have done a great job in the general methodology of the paper and the presentation of their data, however, I have two considerations:

1. There's a problem with the referencing. The references are not arranged in the right order. For example, on line 38, we have references 4 and 5, and then we jump to reference no. 9 on line 39. Also, the authors need to stick to only one referencing style (examples: line 40, 473, 492). Please use a citation manager like EndNote or any free software like Mendeley. 

2. The introduction section is full of ups and downs. Whenever the authors highlight the gap, the aim does not follow it and they circle back to the subject and re-state the gap. I believe the introduction section should end at line 50 and then start again at line 114 through 120. The rest (lines 51-113) can be integrated into the discussion section.

Author Response

Thank you to the reviewer for the comments. We have addressed the comments in the following ways. 

  1. We have corrected the references and referencing format as noted and double-checked all other references. We used Zotero citation software to do the citations initially and have confirmed the software was accurate for this revision.

  1. We have revised the introduction based on all five reviewer comments. We appreciate the suggestion to substantially shorten the introduction. We have considered this in the context of the other sometimes contrary reviewers and hope we have presented a middle ground to best address these comments. We have removed the paragraph discussing qualitative research methods and included this information throughout the methods and discussion sections, as well as deleted a couple of sentences of repeated information once moved to the discussion. We have also added a few sentences throughout the introduction to highlight the research aims in the context of gaps in current literature. The introduction is structured to provide an overview in the initial paragraph, background information of the gaps in the literature that build the argument of the current study in the main body paragraphs, and then the study aims in the final paragraph to clearly highlight the research aims for the current study. We have also reviewed the manuscript for grammatical or spelling errors.

Reviewer 2 Report

Dear Authors,

Your research is very interesting. I read the whole paper with great interest. My comments concern the structure of the article and the issue of the division of the content. While reading the text I had some dissonance because I did not find a division between the introduction and the literature review. The introduction contains various contents that should be found in other parts of the paper.

According to scientific art, the introduction should be general, informing the reader what the text will be about and how it will be divided. It would be good to number the different parts of the text and to inform the reader in the introduction which parts the paper consists of and what they contain (The paper is structured as follows...Section 1... etc.).

All methodological issues should be addressed in the methodology section (not in the introduction). I very much miss the "Literature review", "Theoretic background" section.

In my opinion, after taking these comments into account, the structure of the division of the content of the paper should have been logical and consistent.

My kind regards, Reviewer

Author Response

Thank you for the review of our manuscript. We appreciate your thoughtful comments on how to restructure the introduction and methods. There is likely a difference in our team’s manuscript process that accounts for some of how the current manuscript is written with the suggestions outlined by the reviewer. Specifically, it is not general practice in our work to structure an introduction such that it outlines the structure of the entire manuscript nor to include a specific literature review outside the introduction; both of these differences are likely due to word limits typical of our area. However, we have revised the introduction considering all five reviewer comments to find a middle ground between providing a much more concise introduction and revising/adding additional information. The reviews across reviewers varied dramatically, including some noting no reason for revision. We have provided some revisions to the introductions to improve the clarity of the argument, including removing a section recommended by reviewer and adding further summary statements to support the research aims. The introduction is structured to provide an overview in the initial paragraph, background information of the gaps in the literature that build the argument of the current study in the main body paragraphs, and then the study aims in the final paragraph to clearly highlight the research aims for the current study. Consistent with the above comments, we have moved discussion of qualitative research methods to the methods section and/or discussion section as relevant. 

Reviewer 3 Report

I thought overall this was a well written and clear paper. I only had two minor points.

- The initial mention on p.5 that there were 5 categories was initially a bit confusing since the methodology mentioned 6 (p.4). The explanation for the reduction then does come on p.6 but I wonder if that information would best be moved prior to Table 2 so that the confusion can be addressed sooner.

- The results section ends with #51 (presumably referring to in-text identifiers), where Table 3 only lists 50. Or are these different numbers?

Author Response

Thank you for your review of our manuscript. Please see the responses below to the two points outlined.

1. Thank you for the comment and noting this confusion regarding the methods section. We have revised this section to include the discussion of the changes to the theorized framework prior to Table 2.

2. Thank you for noticing this error. We have corrected the in-text identifier for Table 3. The last two numbers in the text should have read 49 and 50, instead of 50 and 51. 

Reviewer 4 Report

This manuscript addresses a highly topical issue and develops the study in an appropriate manner. However, the introductory section does not clearly describe the aim of the research. In addition, the Discussion section has references in the wrong format. Finally, the Conclusion section should be expanded, it is too short and does not clearly describe the contribution of this research. 

Author Response

Thank you to the reviewer for the comments. We have addressed the comments in the following way. 1) We have revised the introduction to more clearly described the aim of the research, incorporating several comments from other reviewers as well. Specifically, we have moved the paragraph on qualitative research methods to the methods section and discussion section as relevant. We believe this helps to highlight the current paper’s main aims for directly. Further, we have added a few sentences to try to further underscore the aims. 2) We have corrected and double checked all references. 3) We expanded the conclusions section to more clearly describe the contribution of the research. Specifically, we have restated the particular contribution of this research and expanded the key takeaway points, with word limits in mind. We would be happy to expand further if additional information is missing.

Reviewer 5 Report

The manuscript focuses on men perinatal experience under COVID-19. It is an interesting topic and well-discussed. No revision needed. 

Author Response

Thank you to the reviewer for the comments. We have reviewed for spelling/grammar as suggested.

Round 2

Reviewer 1 Report

Thank you for your response. Only one minor comment:

Line 114: please make sure that "yet" is written correctly in the clean version.